# Validation of a Prospective Urinalysis-Based Prediction Model for ICU Resources and Outcome of COVID-19 Disease: A Multicenter Cohort Study

**DOI:** 10.3390/jcm10143049

**Published:** 2021-07-09

**Authors:** Oliver Gross, Onnen Moerer, Thomas Rauen, Jan Böckhaus, Elion Hoxha, Achim Jörres, Matthias Kamm, Amin Elfanish, Wolfram Windisch, Michael Dreher, Juergen Floege, Stefan Kluge, Christian Schmidt-Lauber, Jan-Eric Turner, Samuel Huber, Marylyn M. Addo, Simone Scheithauer, Tim Friede, Gerald S. Braun, Tobias B. Huber, Sabine Blaschke

**Affiliations:** 1Clinic for Nephrology and Rheumatology, University Medical Center Göttingen, 37075 Göttingen, Germany; jan.boeckhaus@med.uni-goettingen.de; 2Clinic of Anaesthesiology, University Medical Center Göttingen, 37075 Göttingen, Germany; omoerer@med.uni-goettingen.de; 3Division of Nephrology and Clinical Immunology, RWTH Aachen University, 52074 Aachen, Germany; trauen@ukaachen.de (T.R.); jfloege@ukaachen.de (J.F.); gbraun@ukaachen.de (G.S.B.); 4III. Department of Medicine, University Medical Center Hamburg-Eppendorf, 20246 Hamburg, Germany; e.hoxha@uke.de (E.H.); christian.schmidt-lauber@uke.de (C.S.-L.); j.turner@uke.de (J.-E.T.); t.huber@uke.de (T.B.H.); 5Department of Medicine I, Nephrology, Transplantation and Medical Intensive Care, University Witten/Herdecke Medical Center Cologne-Merheim, 51109 Cologne, Germany; JoerresA@kliniken-koeln.de (A.J.); drkamm@aol.com (M.K.); ElfanishA@kliniken-koeln.de (A.E.); 6Department of Pneumology and Critical Care Medicine, University Witten/Herdecke, Medical Center Cologne-Merheim, 51109 Cologne, Germany; windischw@kliniken-koeln.de; 7Department of Pneumology and Intensive Care Medicine, Medical Clinic V, RWTH Aachen University, 52074 Aachen, Germany; mdreher@ukaachen.de; 8Department of Intensive Care, University Medical Center Hamburg-Eppendorf, 20246 Hamburg, Germany; skluge@uke.de; 9I. Department of Medicine, University Medical Center Hamburg-Eppendorf, 20246 Hamburg, Germany; s.huber@uke.de (S.H.); m.addo@uke.de (M.M.A.); 10Institute of Infection Control and Infectious Diseases, University Medical Center Göttingen, 37075 Göttingen, Germany; simone.scheithauer@med.uni-goettingen.de; 11Department of Medical Statistics, University Medical Center Göttingen, 37075 Göttingen, Germany; Tim.friede@med.uni-goettingen.de; 12Emergency Department, University Medical Center Göttingen, 37075 Göttingen, Germany; sblasch@gwdg.de

**Keywords:** COVID-19, ICU resources, risk stratification, clinical outcomes, preventive measures, acute kidney injury

## Abstract

In COVID-19, guidelines recommend a urinalysis on hospital admission as SARS-CoV-2 renal tropism, post-mortem, was associated with disease severity and mortality. Following the hypothesis from our pilot study, we now validate an algorithm harnessing urinalysis to predict the outcome and the need for ICU resources on admission to hospital. Patients were screened for urinalysis, serum albumin (SA) and antithrombin III activity (AT-III) obtained prospectively on admission. The risk for an unfavorable course was categorized as (1) “low”, (2) “intermediate” or (3) “high”, depending on (1) normal urinalysis, (2) abnormal urinalysis with SA ≥ 2 g/dL and AT-III ≥ 70%, or (3) abnormal urinalysis with SA or AT-III abnormality. Time to ICU admission or death served as the primary endpoint. Among 223 screened patients, 145 were eligible for enrollment, 43 falling into the low, 84 intermediate, and 18 into high-risk categories. An abnormal urinalysis significantly elevated the risk for ICU admission or death (63.7% vs. 27.9%; HR 2.6; 95%-CI 1.4 to 4.9; *p* = 0.0020) and was 100% in the high-risk group. Having an abnormal urinalysis was associated with mortality, a need for mechanical ventilation, extra-corporeal membrane oxygenation or renal replacement therapy. In conclusion, our data confirm that COVID-19-associated urine abnormalities on admission predict disease aggravation and the need for ICU (ClinicalTrials.gov number NCT04347824).

## 1. Introduction

Coronavirus 2019 disease (COVID-19) is initiated by infection of the upper respiratory tract with the severe acute respiratory syndrome coronavirus 2 (SARS-CoV-2) [1,2]. Major mechanisms of the disease comprise viral replication in multiple organs, including the kidney and involving multiple cell types, vascular endotheliitis, thrombosis, and systemic cytokine storm [3,4,5,6,7,8,9]. While most patients experience mild to moderate disease, a subgroup requires hospitalization. A proportion of this latter group will exhibit a further decline to multiorgan failure requiring mechanical ventilation, extra corporeal membrane oxygenation (ECMO) or renal replacement therapy (RRT) yielding a high mortality [10,11,12,13,14,15,16,17]. Early clinical identification of such patients could improve allocation of medical resources and, thus, the outcome [18]. In a pilot study, we found that early on, abnormalities of the urine, serum albumin and antithrombin III (AT-III) were associated with worse outcomes of COVID-19. Consequently, we proposed the hypothesis in The Lancet that these markers may serve to indicate kidney involvement and loss of important proteins to construct a prediction tool for COVID-19 severity [19]. Here, we present the multicenter cohort validation study (NCT04347824) of this hypothesis.

The rationale for employing urinalysis is supported by our autopsy study demonstrating that renal tropism of SARS-CoV-2 was associated with disease severity, including premature death and with the development of acute kidney injury (AKI) [20]. Tubular, endothelial and interstitial inflammatory damage may all be reflected in the urine in the form of low osmolarity, albuminuria, hematuria and leukocyturia, respectively. In addition, an abnormal urinalysis can also reflect non-viral changes induced by systemic COVID-19 [21]. For practical reasons, here we define both etiologies of urinary abnormalities as ‘COVID-19 associated kidney injury’. Indeed, urinary abnormalities such as protein loss may both indicate and drive severe disease.

The rationale for adding serum albumin and AT-III activity to the novel algorithm derive from our pilot study’s observation that they were particularly low among the sickest COVID-19 patients and that the administration of these substances for substitution in the ICU setting surged during the first COVID-19 wave in Germany [19]. Both parameters are widely used by emergency and intensive care physicians as they play a major role in the pathogenesis of sepsis, and also by nephrologists in nephrotic patients [22,23]. Low serum albumin induces fluid overload, pulmonary edema and circulatory failure, all of which are known to be major causes of death in COVID-19 multiorgan involvement [3,4,5,6,7,8,9]. Low AT-III activity triggers thromboembolic events and counteracts the beneficial effect of heparin.

## 2. Materials and Methods

### 2.1. Data Quality, Quality Assurance and Study Population

In this multi-center cohort study, all patients hospitalized to four tertiary medical centers were screened for inclusion criteria: (1) SARS-CoV-2 diagnosis (by PCR), (2) urinary status during hospital stay, and (3) patient’s consent. Data collection was performed using a standardized, ICH-GCP-conform and pseudonymized questionnaire assessing age, sex, weight, height, information on chronic diseases, urinary status, serum albumin, AT-III activity, need for ICU transfer, need for and time on mechanical ventilation, ECMO, RRT and death. “ICU-admission” was defined as admission to ICU level 3. The study was approved by the leading institutional review board (IRB) of the UMG Göttingen (41/4/20), and all others. Data were analyzed after the SAP (statistical analysis plan) was completed and database closure. “Transparent reporting of a multivariable prediction model for individual prognosis” (TRIPOD) criteria were entirely applied (see TRIPOD Checklist in the Appendix A) as well as the STROBE Statement checklist for cohort studies. The trial was registered prospectively at www.ClinicalTrials.gov (NCT04347824).

### 2.2. Urinalysis and Patient Stratification

An abnormal urinary status was defined as either anuria or as at least two of the following criteria: (1) urine osmolarity/specific gravity below normal values; (2) leukocyturia; (3) hematuria; (4) albuminuria/proteinuria. If urine was positive for nitrite or bacteria, at least three of these criteria were necessary to define an abnormal urinary status. Serum albumin and AT-III activity were assessed by an enzymatic color test with photometric determination using (1) for serum albumin: Abbott Architect C-Module (Abbott Park, IL, USA), Siemens Atellica CH (Erlangen, Germany), Roche cobas c 502 (Rotkreuz, Switzerland), Beckman Coulter AU Clinical Chemistry Analyzer (Krefeld, Germany); and (2) for AT-III activity: Instrumentation Laboratory Werfen ACL TOP 750 (Munich, Germany), Siemens Atellica COAG (Erlangen, Germany), Instrumentation Laboratory ACL TOP (Bedford, MA, USA), Siemens BCS XP (Erlangen, Germany).

Eligible patients were then stratified into three predefined groups as follows: (1) low-risk for COVID-19 kidney injury (“green”)—patients without urinary abnormalities; (2) intermediate-risk (“yellow”)—patients with an abnormal urinary status, serum albumin ≥ 2.0 g/dL and AT-III activity ≥ 70%; (3) high-risk (“red”)—patients with an abnormal urinary status, serum albumin < 2.0 g/dL or AT-III activity < 70%.

### 2.3. Study Endpoints and Statistical Analyses

The primary endpoint was time to event (in days) defined as time to ICU transfer or death, whatever occurred first, within the first ten days upon admission. Secondary endpoints included the number of patients who died, were transferred to ICU or required mechanical ventilation, ECMO or RRT.

As per study protocol (see Appendix A), the aim was to recruit 100 to 250 patients. The primary comparison was between low-risk and intermediate/high-risk patients with regard to the primary endpoint. The expectation was that only 20% of patients will be in the low-risk group whereas 80% will be in the intermediate or high-risk groups. The risk for a primary outcome event was considered to be ~25% (at day 10) in the low-risk group and ~50% in the composite intermediate/high-risk group with a relative risk (RR) of 2. Under these assumptions, a total sample size of 183 or 240 patients yielded a power of 80% or 90%, respectively, at the typical two-sided significance level of 5%. When postulating a risk of 30% in the low-risk group and otherwise the same assumptions (in particular RR = 2), a total sample size of 130 (172) patients yielded a power of 80% (90%).

To assess the association of the risk groups as determined by the novel algorithm [19], a frequency table of the risk groups and the type of ward are provided. The association was formally tested by a chi-square test.

The primary endpoint was analyzed using a Cox proportional hazards model with the risk groups as the independent variable (see Appendix A). The primary null hypothesis was that the hazard ratio (HR) for low-risk vs. intermediate/high-risk is higher or equal to one, i.e., H0:HR ≥ 1. The hazard ratio is reported with a 95% confidence interval (CI) and *p*-value (Wald-type test). The performance of the prognostic model was evaluated using the C-index and calibration slope (reported with standard errors (SE)). In supporting analyses, additional prognostic variables were added to the models. These include age, sex, presence of any comorbidity (e.g., kidney function, diabetes, cardiovascular disease, chronic respiratory disease).

## 3. Results

Patients, who were transferred from the external ICU to the University ICU (*N* = 14) and patients with a urinary status older than 48 h following admission (*N* = 64), were excluded from prediction analysis and included to the association analysis, which is shown in Table 1. Among the entire cohort of 223 patients, 145 had a urinalysis prospectively obtained and thereby qualified for the validation analysis, whereas 78 patients had a urinalysis that was obtained later during hospital stay (*N* = 64) or were secondary referrals from ICU of other medical centers (*N* = 14; Figure 1).

When looking at the entire cohort, we found similar baseline characteristics of both groups and could confirm the overall association of urine abnormalities with dismal outcomes in a statistically significant fashion (see Table 2 for details).

Based on the proposed algorithm derived from our pilot study [19], our major goal was to validate the predictive value of this algorithm in those 145 COVID-19 patients who had a prospective urinalysis. Based on our pre-specified diagnostic classification criteria (see methods), 43 individuals (30%) had a normal (categorized to “green”, low-risk) and 102 (70%) had an abnormal urinalysis with the latter group sub-differentiating into 84 intermediate-risk (“yellow”) and 18 high-risk (“red”) patients (Figure 2, Table 2). Patients with an abnormal urinalysis at hospitalization had a higher risk for ICU transfer or death (*N* = 65/102, 63.7%) as compared to those without urinary abnormalities (*N* = 12/43, 27.9%; HR 2.6; 95%-CI 1.4 to 4.9; *p* = 0.0020; Table 2 and Table 3, Figure 2).

The HR remained stable even after adjusting for age, sex and co-morbidities (Table 4) with an HR of 2.6, a 95% CI of 1.4 to 4.9 and a *p* = 0.0020. Based on the risk stratification, all patients in the high-risk group, 51.2% of the intermediate-risk group and only 23.3% of the low-risk group reached the primary composite endpoint (Figure 2, Table 3). The C-index (SE) for this simple Cox regression was 0.61 (0.03). The adjustment of the prediction model for age, sex and co-morbidities revealed a robust hazard ratio for COVID-19 kidney injury, which was consistent with the unadjusted hazard ratio (C-index 0.68 (0.04)) (Table 4). Patients with urinary abnormalities on admission to hospital had a higher in-hospital mortality (29.4% vs. 11.6%), a higher risk to need mechanical ventilation (44.1% vs. 14.0%), ECMO therapy (10.8% vs. 2.3%) and RRT (30.7% vs. 11.6%; Table 3). Primary and secondary outcome events were particularly elevated in high-risk patients as compared to those at intermediate risk (Table 3, Figure 2). In parallel, despite death as a competing event, the risk groups also correlated with the average patient time in the ICU and for days on mechanical ventilation, ECMO-therapy, and RRT (Table 3), which might be of special interest in scenarios of limited resources and limited ICU capacities [10].

The prospectively chosen categorical ranges for serum albumin and AT-III served to sub-stratify the intermediate and high-risk groups. To further analyze these parameters as hypothetic drivers of COVID-19, we plotted the earliest available and lowest serum albumin and AT-III levels of the three risk groups (Figure 3). Median values of these parameters declined with the traffic light coding system from “green” to “red”, thereby supporting the definitions used. The delta between earliest and lowest value was surprisingly stark in the intermediate-risk group for serum albumin and in the high-risk group for AT-III activity. This might reflect the dynamics of the disease within these categories. While only 3.0% of low-risk patients had a minimal serum albumin below 2.0 g/dL, this was noted in 30.0% and 94.4% of the intermediate and high-risk groups, respectively. Similarly, 22.7% of the low-risk, 40.4% of the intermediate-risk and 66.7% of the high-risk group reached minimum levels of AT-III lower than 70%. This translates to almost 50% of intermediate-risk patients deteriorating from “yellow” to the “red” category during their hospital stay, thus, implying a dynamic change in disease severity.

## 4. Discussion

The present validation study confirmed that our easily accessible algorithm predicts the severity and the further course of COVID-19 disease on hospital admission, which also includes the need for ICU resources.

The majority of patients were in the intermediate risk group. Notably, 30% of this group showed a decline in their serum albumin to the high-risk category during their hospital stay and AT-III activity fell in 40% below the critical cut-off of 70% (Figure 3). Low AT-III activity is known to trigger thromboembolic complications. Preventive measures require a much more aggressive dosing of heparin, substitution of AT-III, or the therapeutic switch to another anticoagulant that does not work via AT-III, as discussed in the recent clinical practice guideline [24]. The C-index of 0.61 is not indicative for a high accuracy for the model, however, if one takes into account that a simple urinalysis still has such a predictive accuracy, this finding is very helpful in the rather confused setting of many emergency departments. Our algorithm indicates that those with normal urine have a substantially lower risk progressing to ICU or death. Yet, the algorithm puts a high number of patients in the focus: this group might benefit most from our algorithm as preventive measures against multiorgan disease can start earlier, as recommended in the German guideline [24].

At present, it is unclear which pathomechanisms contribute most to the low albumin and low AT-III activity. One may speculate that low AT-III activity is caused by an increased consumption due to a hypercoagulable state, increased loss by capillary leak, decreased hepatic production and, importantly, renal loss. COVID-19 patients exhibit frequent thromboembolic events, and pulmonary embolism has evolved as a major killer in COVID-19 [25,26,27,28]. Low levels of AT-III trigger this complication and counteract the effect of the most common anticoagulant heparin. In addition, patients with COVID-19 show pulmonary interstitial edema due to severe fluid overload, and circulatory failure [22]. In the latter setting, low serum albumin triggers this complication and is believed to be the result of increased loss by capillary leak and decreased hepatic production. Low serum albumin may also explain the reduced response to some medications by impaired plasma protein binding.

The first limitation relates to the use of non-interventional data from clinical routine which has also led to the loss of data points in some patients and the partial use of retrospective data, which can be explained by the extreme circumstances of a pandemic [29]. However, the study design has a clear prospective nature, as our key inclusion criteria was the predictive urinalysis obtained prior to the event. Notably, median serum creatinine on hospital admission (Table 2) did not differ in between the low-risk group and the intermediate/high-risk group, which underlines the value of the urinalysis for our prediction model. As the second limitation, despite prospective urinalysis prior reaching the primary endpoint “ICU-admission”, it is impossible to discriminate from urine alone between SARS-CoV-2 infection of the kidney, ICU-related acute kidney injury, cytokine storm or additional confounders [30]. However, discrimination between acute or chronic renal injury is less important in the triage room, where the highest need of the COVID-19 patient is a fast and easy to use risk-assessment [31]. As the third limitation, the small sample size, co-morbidities (Table 4) and additional diseases might be confounding factors for the need for ICU and mortality. Ten percent of patients worldwide have pre-existing CKD, and urinary tract infections or indwelling catheters represent a potential confounder. As the fourth limitation, all four participating trial centers used the same photometric method for the determination of serum albumin and AT-III activity. However, suppliers of the enzymatic color tests differed as did the laboratory machines, which carries a risk of test variation in between the different trial centers. Lastly, a limitation lies in the early recordings of albumin and AT-III on hospital admission, whereas the disease is highly dynamic, thus, inherently compromising on the predictive power as compared to repeated measurements. As one strength, our study’s C-index, which is indicative of the accuracy of a prediction model, was 0.61. This value suggests our prediction model as being accurate and useful in the real-life setting, since it stayed significant despite potential negative influences. However, thorough clinical assessment is still needed in all patients with COVID-19 disease upon arrival in the emergency room, as many other negative influences and underlying diseases also can influence their outcome and need for ICU resources.

## 5. Conclusions

Our simple algorithm provides a strategic advantage when it comes to the triage and surveillance of COVID-19 patients seeking hospital treatment. The algorithm can also be used in a modified way for better surveillance of outpatients and more vulnerable populations such as elderly patients in nursing homes. We advocate its use for better allocation of patients at risk or with the most need for specific therapies [32,33,34,35,36]. Furthermore, as preventive measures will improve over time, the algorithm can be adapted with updated recommendations [24].

## Figures and Tables

**Figure 1 jcm-10-03049-f001:**
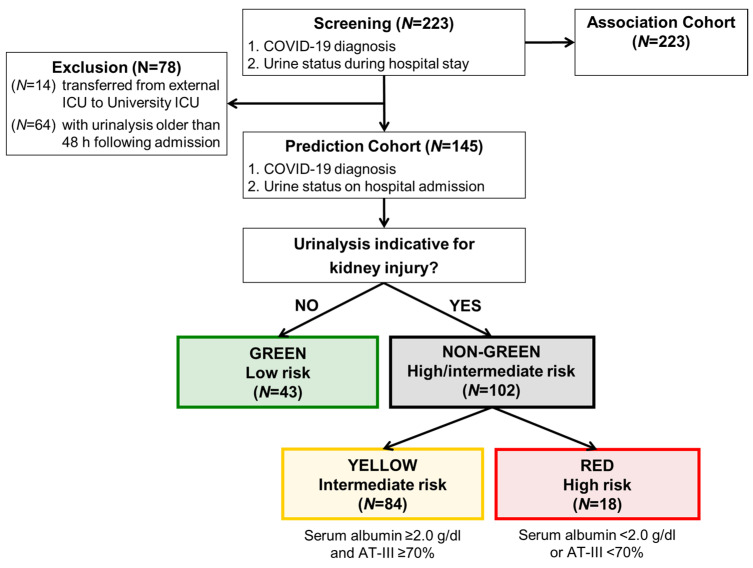
Algorithm for prediction model development (study flow chart). Patients with a urinary status on admission to hospital were allocated to low, intermediate and high-risk. The results of the prediction analysis are shown in Table 2 and Table 3 and Figure 2 and Figure 3. The first event within 10 days was counted, admission to ICU or death, whatever came first. ^1^ Patients, who initially served for generation of the algorithm were not part of this cohort [19]. ^2^ Patients, who were transferred from external ICU to University ICU (*N* = 14) and patients with urinary status older than 48 h following admission (*N* = 64) were excluded from prediction analysis and included to the association analysis, which is shown in Table 1. ICU—intensive care unit; AT-III—antithrombin III activity.

**Figure 2 jcm-10-03049-f002:**
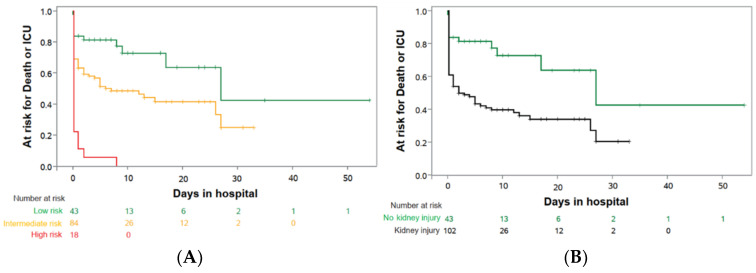
Prediction of “death or need for ICU” on entry to hospital. Kaplan–Meier estimates prediction analysis for patients with urinary status on admission to hospital: (**A**) patients without kidney injury (low-risk; green; *N* = 43) vs. patients with kidney injury, divided in intermediate-risk (yellow; *N* = 84) and high-risk (red; *N* = 18); (**B**) patients with kidney injury (non-green; *N* = 102) vs. patients without kidney injury (normal urine status; green; *N* = 43).

**Figure 3 jcm-10-03049-f003:**
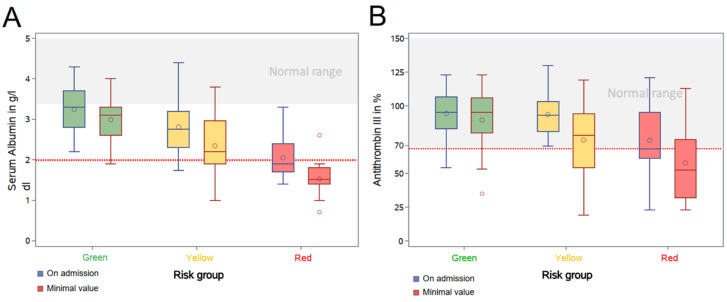
Differences and development of serum albumin and antithrombin III of the different cohorts (low, intermediate and high-risk) within the prediction analysis. (**A**) Serum albumin on hospital admission and minimum level during hospital stay; (**B**) Antithrombin III (AT-III) activity on hospital admission and minimum level during hospital stay. Both, serum albumin and AT-III illustrate differences in between low-risk (green) and intermediate or high-risk (yellow and red). Remarkably, a substantial number of patients in the intermediate-risk group, but not the low-risk group, exhibited a worsening course of their hospital stay, reflecting the possible benefits of our algorithm as an early warning system in COVID-19 systemic disease.

**Table 1 jcm-10-03049-t001:** Association (all patients with a urine status). Patients’ characteristics of individuals with a urine status, including the primary endpoint and secondary endpoints. ICR—interquartile range; ICU—intensive care unit (level 3); ECMO—extra-corporal membrane oxygenation; RRT—renal replacement therapy; BMI—body mass index.

Characteristic	Association: Urine Status in Relation to Disease Severity (*N* = 223)				
	Green (*N* = 58)		Non-Green (*N* = 165)		Green vs. Non-Green
**Male sex**—no. (%)	40/58	(69)	114/165	(69.1)	0.98
**Median age (IQR)**—year	60	(25)	66	(19)	<0.01
**Median body mass index BMI (IQR)**	27.2	(7.5)	27.4	(6.9)	0.778
**Chronic disease PRIOR to COVID-19**—no. (%)	44/58	(75.9)	122/165	(73.9)	
Chronic heart disease—no. (%)	16/58	(27.6)	46/165	(27.9)	
Chronic lung disease—no. (%)	16/58	(27.6)	28/165	(17)	
Chronic kidney disease—no. (%)	6/58	(10.3)	17/165	(10.3)	
Renal replacement therapy—no. (%)	1/58	(1.7)	4/165	(2.4)	
Malignant tumor disease—no. (%)	13/58	(22.4)	14/165	(8.5)	
Diabetes mellitus—no. (%)	8/58	(13.8)	35/165	(21.2)	
Immunosuppressive therapy—no. (%)	19/58	(32.8)	29/165	(17.6)	
**Median serum albumin (IQR)**—g/dL					
First value in hospital (IQR)—g/dL	3.3	(0.9)	2.4	(0.9)	
Lowest value (IQR)—g/dL	3	(0.9)	2	(0.8)	
<2.0 g/dL during hospital stay—no. (%)	3/47	(6.4)	70/148	(47.3)	
**Median antithrombin III (IQR)**—%					
First value in hospital (IQR)—%	92.5	(23)	85	28	
Lowest value (IQR)—%	90	(30)	63	37	
<70% during hospital stay—no. (%)	7/28	(25)	67/118	(56.8)	
**ICU or death until day 10**—no. (%)	**14/58**	**(24.1)**	**109/164**	**(66.5)**	**<0.01**
**Death during hospital stay**—no. (%)	7/58	(12.1)	47/165	(28.5)	<0.01
Still at risk of death in hospital—no. (%)	2/58	(3.4)	30/165	(18.2)	
**ICU level 3 during hospital stay**—no. (%)	13/58	(22.4)	106/165	(64.2)	
**Mechanical ventilation**—no. (%)	9/58	(15.5)	95/165	(57.6)	<0.01
**ECMO therapy**—no. (%)	1/58	(1.7)	31/164	(18.9)	<0.01
**Renal replacement therapy RRT**—no. (%)	7/58	(12.1)	65/164	(39.6)	<0.01

**Table 2 jcm-10-03049-t002:** Baseline characteristics (urine status on admission to hospital). Patients’ characteristics of individuals. IQR—interquartile range; BMI—body mass index.

Baseline Characteristic	Green (*N* = 43)		Yellow (*N* = 84)		Red (*N* = 18)		Non-Green (*N* = 102)	
**Male sex**—no. (%)	29/43	(67.4)	53/84	(63.1)	12/18	(66.7)	65/102	(63.7)
**Median serum creatinine (IQR)**—mg/dL	0.97	(0.97–1.18)					1.0	(0.76-1.63)
**Median age (IQR)**—year	61	(49-75)	71	(60-79)	67	(58-74)	70	(59-78)
20 to 39 years of age	8/43	(18.6)	4/84	(4.8)	0/18	(0)	4/102	(3.9)
40 to 59 years of age	12/43	(27.9)	17/84	(20.2)	5/18	(27.8)	22/102	(21.6)
60 to 79 years of age	20/43	(46.5)	44/84	(52.4)	11/18	(61.1)	55/102	(53.9)
80 and older years of age	3/43	(7)	19/84	(22.6)	2/18	(11.1)	21/102	(20.6)
**Median body mass index BMI (IQR)**	27	(24-31)	26	(24-31)	27	(26-33)	26	(24-31)
**Chronic disease PRIOR COVID-19**—no. (%)	31/43	(72.1)	73/84	(86.9)	12/18	(66.7)	85/102	(83.3)
Chronic heart disease—no. (%)	10/43	(23.3)	28/84	(33.3)	6/18	(33.3)	34/102	(33.3)
Chronic lung disease—no. (%)	13/43	(30.2)	15/84	(17.9)	5/18	(27.8)	20/102	(19.6)
Chronic kidney disease—no. (%)	4/43	(9.3)	16/84	(19)	0/18	(0)	16/102	(15.7)
Renal replacement therapy—no. (%)	1/43	(2.3)	4/84	(4.8)	0/18	(0)	4/102	(3.9)
Malignant tumor disease—no. (%)	8/43	(18.6)	12/84	(14.3)	1/18	(5.6)	13/102	(12.7)
Diabetes mellitus—no. (%)	4/43	(9.3)	23/84	(27.4)	6/18	(33.3)	29/102	(28.4)
Immunosuppressive therapy—no. (%)	13/43	(30.2)	22/84	(26.2)	3/18	(33.3)	25/102	(24.5)

**Table 3 jcm-10-03049-t003:** Primary and secondary outcome measures (urine status on admission to hospital). Patients’ outcome measures, including the primary endpoint and secondary endpoints. IQR—interquartile range; ICU—intensive care unit (level 3); ECMO—extra-corporal membrane oxygenation; RRT—renal replacement therapy; DPP—days per patient (estimate of patient days on mechanical ventilation, ECMO or RRT per patient in each group).

Outcome	Green (*N* = 43)		Yellow (*N* = 84)		Red (*N* = 18)		Non-Green (*N* = 102)	
**Primary endpoint**								
**ICU or death until day 10—no. (%)**	**10/43**	**(23.3)**	**43/84**	**(51.2)**	**18/18**	**(100.0)**	**61/102**	**(59.8)**
**Secondary endpoints—Complications**								
**Death during hospital stay**—no. (%)	5/43	(11.6)	23/84	(27.4)	7/18	(38.8)	30/102	(29.4)
Still at risk of death in hospital—no. (%)	1/43	(2.3)	8/84	(9.5)	3/18	(16.7)	11/102	(10.8)
**ICU level 3 during hospital stay**—no. (%)	9/43	(20.9)	38/84	(45.2)	17/18	(94.4)	55/102	(53.9)
**Mechanical ventilation**—no. (%)	6/43	(14)	28/84	(33.3)	17/18	(94.4)	45/102	(44.1)
**ECMO therapy**—no. (%)	1/43	(2.3)	5/84	(6.0)	6/18	(33.3)	11/102	(10.8)
**Renal replacement therapy RRT**—no. (%)	5/43	(11.6)	20/83	(24.1)	11/18	(61.1)	31/101	(30.7)
**Secondary endpoints—Resources**								
**Time on mechanical ventilation**—days (DPP)	23	(2.6)	23	(7.7)	10	(9.4)	19	(8.4)
**Time on ECMO therapy**—days (DPP)	11	(0.3)	20	(1.2)	23	(7.7)	22	(2.3)
**Time on RRT**—days (DPP)	23	(2.7)	17	(4.1)	13	(7.9)	15	(4.6)
**Secondary endpoints—Blood values**								
**Median serum albumin (IQR)**—g/dL								
First value in hospital (IQR)—g/dL	3.3	(0.9)	2.8	(0.9)	1.9	(0.7)	2.7	(1)
Lowest value (IQR)—g/dL	3.1	(0.7)	2.2	(1.1)	1.5	(0.4)	2.1	(1.1)
<2.0 g/dL during hospital stay—no. (%)	1/33	(3.3)	21/70	(30)	17/18	(94.4)	38/88	(43.2)
**Median antithrombin III (IQR)**—%								
First value in hospital (IQR)—%	95	(23.5)	93	(22)	68	(34)	89	(24.5)
Lowest value (IQR)—%	95	(26)	78	(40)	52.5	(43)	71	(42)
<70% during hospital stay—no. (%)	5/22	(22.7)	21/52	(40.4)	12/18	(66.7)	33/70	(47.1)

**Table 4 jcm-10-03049-t004:** Adjustment of the COVID-19 kidney injury prediction model for age, sex and co-morbidities. Adjustment revealed in a robust hazard ratio for COVID-19 kidney injury, which was consistent with the unadjusted hazard ratio. Only five patients were on chronic RRT prior to COVID-19 infection, resulting in a broad range in the 95% confidence interval and a hazard ratio of 3.1, which just failed to reach the level of significance. RRT—renal replacement therapy.

Parameter	Hazard Ratio	95% Confidence Intervals		*p*
**Kidney injury**	2.868	1.484	5.505	0.0017
**Age (year)**	0.995	0.979	1.011	0.5412
**Sex: male**	1.088	0.660	1.794	0.7417
**Chronic heart disease**	1.191	0.673	2.110	0.5480
**Chronic lung disease**	1.311	0.756	2.273	0.3355
**Chronic kidney disease**	0.645	0.275	1.513	0.3138
**RRT**	3.132	0.897	10.938	0.0736
**Malignant tumor disease**	0.888	0.337	2.344	0.8109
**Diabetes mellitus**	0.973	0.557	1.701	0.9247
**Immunosuppressive therapy**	0.938	0.444	1.980	0.8660

## Data Availability

The full trial protocol and the statistical analysis plan can be accessed as Appendix A. The full dataset can be assessed on individual request through a collaborative process. Please contact gross.oliver@med.uni-goettingen.de for additional information.

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
