# Peer review of "Validation of a Prospective Urinalysis-Based Prediction Model for ICU Resources and Outcome of COVID-19 Disease: A Multicenter Cohort Study"

_jcm, 2021, doi:10.3390/jcm10143049_

Round 1

Reviewer 1 Report

The authors highlight the role of urinalysis in COVID-19.

Unfortunately, the reader is not informed about the methodology used, making review of the reported findings very difficult.

1.The albumin is assayed. HOW was this achieved? (bromocresol green-based dye binding method? bromocresol purple method? (this is of major importance since this choice determines the reactivity of carbamylated albumin and alpha 2 macroglobulin in these assays), nephelometric/turbidimetric method? electrophoresis? Was the method comparable in the various participating centres? the reader has to guess.

This lack of description does not allow to interprete results: the acute phase proteins alpha 1 acid glycoprotein and alpha 2 macroglobulin are partially detected by this assay. As in COVID-19 a marked acute phase response takes place, it is likely that the results could be explained by these unspecific reactions against the dye used. This fact has been completely ignored by the authors.

In case of "kidney injury" (observed by the authors) carbamoylated albumin is formed, causing problems to the colorimetric albumin assay (to a variable extent - the reader cannnot judge because the method is not mentioned). To what extent does carbamoylated albumin differences explain the differences - this fact has also been ignored by the authors.

Another problem is the ill-defined "urinalysis". How were the results interpreted. What is the cut-off level used? Is the "urinalysis" based on test strip reading, flow cytometry, automated pattern recognition , counting chamber testing or manual analysis. Again the reader is uninformed. What were the cut-off criteria used? How were the obtained results corrected for urinary dilution?

As nitrite is reported, we assume that a strip might have been used. Nitrate can only be formed when a lot of nitrate (originating from the diet) . Does the central role of nitrite in the decision tree used imply that patients are on a nitrate containing diet (rich in vegetables) while staying in the intensive care unit?

Reviewer 2 Report

The authors present an analysis aiming to validate the use of abnormal urinalysis to predict poor outcomes following hospital admission with COVID-19. There is a clear a priori hypothesis and the analysis has been carried out according to a pre-specific SAP. The proposed usage of more easily available urinalysis for prognostication and treatment planning is of value. There are however a number of suggestions which I believe would improve the analysis and manuscript prior to publication.

Please clarify the timepoints for the assessed tests, if urinalysis, serum albumin and antithrombin-III all measured within 48 hours of hospital admission. If so, in the discussion (page 8, line 259), ICU-related AKI should not relate to urinalysis on admission. If these were taken at different timepoints, it would be important to know more about their trajectories over the duration of hospital admission and better understand how they relate to each other.

As detailed in the discussion, the above measures likely relate to AKI, fluid overload/circulatory failure, thromboembolic events respectively. It would be very valuable to know what the rates of these events meeting clinical diagnosis were in each outcome category. How does ‘non-green’ kidney injury fit with CKD-EPI or biochemistry definitions for AKI? As these events have been shown to be independently related to poor outcomes in COVID-19, this study will add more to current knowledge if it potentially identifies patients with subclinical disease. Particularly as a C-index of 0.61 does not arguably indicate a high accuracy for the model.

The baseline characteristics for different chronic diseases are actually very different between outcome groups e.g. chronic lung disease, malignant tumour disease, immunosuppressive therapy much higher in the ‘green’ and RRT, DM in the ‘non-green’ groups. We know that these conditions independently increase risk of poor outcomes in COVID-19. Please comment on the implications of this for this analysis. Could these factors be skewing the risk of need for ICU therapy and death?

Unadjusted HRs should be presented alongside adjusted HRs in Table 4 and discussed accordingly.

Please clarify the definition of ICU admission in the primary endpoint vs ICU level 3 admission.

The small sample size particularly in intermediate and high-risk groups needs to be acknowledged as a limitation.

The TRIPOD checklist and STROBE statement appear to be missing from supplement.

Round 2

Reviewer 1 Report

the science remains weak with ill defined methods and criteria

the reader even has to read that albumin is measured using an "enzymatic colorimetric test" (which does not exist ......)

one does not realise that colorimetric albumin is hampered by bilirubin levels (bilirubin and the dye compete for the same binding position of albumin)